# Mediterranean Diet: The Beneficial Effects of Lycopene in Non-Alcoholic Fatty Liver Disease

**DOI:** 10.3390/jcm11123477

**Published:** 2022-06-16

**Authors:** Ludovico Abenavoli, Anna Caterina Procopio, Maria Rosaria Paravati, Giosuè Costa, Nataša Milić, Stefano Alcaro, Francesco Luzza

**Affiliations:** 1Department of Health Sciences, University “Magna Graecia”, 88100 Catanzaro, Italy; procopioannacaterina@unicz.it (A.C.P.); mrparavati@unicz.it (M.R.P.); gcosta@unicz.it (G.C.); alcaro@unicz.it (S.A.); luzza@unicz.it (F.L.); 2Net4Science Academic Spin-Off, University “Magna Graecia”, 88100 Catanzaro, Italy; 3Associazione CRISEA—Centro di Ricerca e Servizi Avanzati per l’Innovazione Rurale, 88055 Belcastro, Italy; 4Department of Pharmacy, Faculty of Medicine, University of Novi Sad, Hajduk Veljkova 3, 21000 Novi Sad, Serbia; natasa.milic@mf.uns.ac.rs

**Keywords:** lycopene, non-alcoholic fatty liver disease, Mediterranean diet, polyphenols, antioxidants activity, nutraceutics, natural compounds

## Abstract

Non-alcoholic fatty liver disease (NAFLD) presents the most common chronic liver disease globally; it is estimated that 25.24% of the world’s population has NAFLD. NAFLD is a multi-factorial disease whose development involves various processes, such as insulin resistance, lipotoxicity, inflammation, cytokine imbalance, the activation of innate immunity, microbiota and environmental and genetic factors. Numerous clinical studies have shown that the Mediterranean diet produces beneficial effects in NAFLD patients. The aim of this review is to summarize the beneficial effects of lycopene, a soluble pigment found in fruit and vegetables, in NAFLD.

## 1. Introduction

Non-alcoholic fatty liver disease (NAFLD) presents the most common chronic liver disease globally. It is estimated that 25.24% of the world’s population has NAFLD with the highest prevalence rates in the Middle East and South America [1]. NAFLD is a multifactorial disease whose development involves various processes, such as the insulin resistance, lipotoxicity, inflammation, cytokine imbalance, the activation of innate immunity, microbiota and environmental and genetic factors [2,3]. Nutrition plays a fundamental role among many factors that can determine the development of NAFLD. In fact, the protracted increase in caloric intake is the critical point for the development of steatosis. As it has already been indicated several times, a high-fat diet (HFD) is one of the main causes of NAFLD as it causes an increase in the concentration of free fatty acids in the liver resulting in steatosis. Moreover, genetic factors are important in the development of NAFLD. In fact, it has been demonstrated that a certain genetic background is associated with its development [4]. One of the most studied genes involved in NAFLD is patatin-like phospholipase 3 (PNPLA3), particularly its single nucleotide polymorphism I148M. The PNPLA3 gene encodes a protein named adiponutrin, which has significant homologies to enzymes implicated in lipid metabolism processes and could exert a lipolytic activity on triglycerides (TG) [5]. In patients with PNPLA3 1148M polymorphism, it has been found that hepatic fat accumulation is twofold higher than in non-carrier patients [6]. Moreover, insulin resistance is an important factor generally involved in the development of NAFLD. Lipid homeostasis is regulated by insulin, which controls the lipolysis of TG when insulin resistance occurs and insulin signaling is compromised, causing dysregulated lipolysis and resulting in the excessive delivery of fatty acids to the liver [7]. Recently, it has been shown that the adipose tissue is able to secrete hormones called adipokines, such as leptin and adiponectin [8]. It has been observed that leptin levels were increased in obese patients as compared to the controls, while conversely, the levels of adiponectin, a hormone with anti-inflammatory and hepato-protective activities, were reduced, resulting in hepatic steatosis and the activation of inflammation and fibrogenesis [9]. More generally, the adipose tissue contributes to the maintenance of low-grade inflammatory states by producing pro-inflammatory cytokines. Furthermore, an increased expression of inflammatory genes and macrophage activation in the visceral and subcutaneous adipose tissue in NAFLD patients correlates with the progression from simple steatosis to non-alcoholic steatohepatitis (NASH) and fibrosis [10,11].

In NAFLD pathogenesis, mitochondrial dysfunction is characterized by structural alterations, such as the depletion of mitochondrial DNA and morphological and ultra-structural changes and functional alterations, involving the respiratory chain and mitochondrial β-oxidations [12]. In fact, when lipid flux is increased, the mitochondrial and peroxisomal function cannot handle it. In this case, respiratory oxidation could collapse, involving the impairment of fat homeostasis, the generation of lipid-derived toxic metabolites and the overproduction of reactive oxygen species (ROS) [13]. Particularly, the formation of lipotoxic species is the event that leads to hepatocellular stress and injury, fibrogenesis, NASH and, finally, cirrhosis and hepatocarcinoma (HCC) [14]. Currently, the specific lipotoxic species that induce cell injury have not been identified, but diacylglycerols (DAG) and ceramides could be involved [15,16,17,18,19]. Additionally, DAG and ceramides could induce hepatic insulin resistance, functioning as intermediaries between lipotoxity and insulin resistance [20,21].

## 2. Mediterranean Diet and NAFLD

The Mediterranean diet (MD) is a complete food model characterized by a wide variety of food products. The main characteristics of MD are the low consumption of saturated fat and cholesterol, high consumption of monounsaturated fatty acid (MUFA), a balanced omega-6 to omega-3 polyunsaturated fatty acids (PUFA) ratio and a high content of complex carbohydrates and fibres [22]. Many clinical studies have shown that MD produces beneficial effects in NAFLD patients. In a study of 73 adult NAFLD patients adhering to MD, the estimated MedDiet Score was associated with lower fatty liver disease and a reduced degree of insulin resistance [23]. In a study conducted on NAFLD patients, the effects of MD on steatosis and insulin sensitivity were examined; adherence to MD showed a significant reduction in steatosis [24]. In a study conducted on a sample of 82 patients, the relations between the adherence to MD and the histological characteristics of NAFLD patients were analysed, indicating that a greater adherence to MD was associated with a lower likelihood of high-grade steatosis and the presence of steatohepatitis [25]. A prospective study conducted on 50 overweight Caucasian patients, randomized into three groups (groups A, B and C), evaluated the beneficial effects of the treatment with MD alone or in combination with antioxidants [26]. The subjects of group A were treated with a moderately low-calorie MD, the subjects of group B were prescribed a low-calorie MD with an antioxidant supplementation and the subjects of group C did not receive any type of treatment. The study showed that MD alone or in combination with antioxidants improved anthropometric parameters, lipid profile, reduced hepatic fat accumulation and hepatic stiffness. Furthermore, the patients in group B showed a significant improvement in insulin sensitivity and a reduction in anthropometric parameters compared with the patients in group A. In a study involving 46 adults with NAFLD, the effect of a clinical intervention and a diet based on MD lasting 6 months was evaluated [27]. The results of the study indicated that the percentage of patients with steatosis grade 2 or greater was reduced from 93% to 48% and steatosis regressed in nine patients (20%). At the end of the treatment, the weight-related end-point, presented by a 7% weight reduction or achievement/maintenance of normal weight, was observed in 25 patients (54.3%). During the treatment, the levels of liver enzymes decreased; the reduction was evident for the alanine aminotransferase (ALT) enzyme with the decrease being from 67% to 11%. The parameters of BMI, waist circumference, aspartate aminotransferase (AST), gamma-glutamyl transferase (GGT), high-density lipoprotein (HDL), serum glucose, Tot-Chol/HDL, low-density lipoprotein LDL/HDL, homeostatic model assessment (HOMA), fatty liver index (FLI), Kotronen index and NAFLD liver fat score showed significant improvements (*p* < 0.01) between the baseline and the end of treatment. In a study of non-diabetic patients with NAFLD, the subjects underwent 16 weeks of MD, 16 weeks of washout and 16 weeks of low-fat diet [28]. At the end of 16 weeks of MD, a significant reduction in mean body weight (−5.3 ± 4.1 kg, *p* = 0.003), mean waist circumference (−7.9 ± 4.9 cm, *p* = 0.001), mean ALT levels (−28.3 ± 11.9 IU/L, *p* = 0.0001) and AST (−6.4 ± 56.3 IU/L, *p* = 0.01) were observed. In a study conducted by Franco et al., 144 patients with moderate to severe NAFLD were randomized into six groups and treated as follows: (1) control diet; (2) low glycemic index MD (LGIMD); (3) aerobic physical activity program (PA1); (4) combined program of physical activity (aerobic activity and resistance training) (PA2); (5) LGIMD plus PA1; and (6) LGIMD plus PA2 [29]. The study results indicated a statistically significant reduction in NAFLD score after 45 days of treatment in every treated group except in group 1. In a study conducted by Ristic-Medic et al., 24 overweight or moderately obese men were causally assigned to two groups and treated with MD or a low-fat diet, respectively [30]. After the treatment, all participants had a significant weight loss (>9%) with improvements in waist circumference and visceral adiposity index (VAI). In addition, the subjects treated with MD had higher levels of HDL cholesterol, lower levels of saturated fatty acids and TG than the subjects treated with a low-fat diet. In a study conducted by Properzi et al., the effects of two isocaloric ad libitum diets (MD or low-fat) on fatty liver and cardiometabolic risk factors were evaluated for 12 weeks [31]. At week 12, hepatic steatosis was significantly reduced in both groups (*p* < 0.01) and there was no difference in liver fat reduction between the groups (*p* = 0.32). In a study conducted by Della Corte et al., the relation between MD and NAFLD was evaluated [32] in a group of children and adolescents with obesity. The level of adherence to MD was assessed using a clinical questionnaire, the quality index of the Mediterranean diet for children and adolescents (KIDMED). The study results indicated that the prevalence of a low KIDMED score was significantly higher in patients with NASH. Moreover, higher C-reactive protein and fasting insulin values were observed in the patients with poor MD adherence.

Another important characteristic of MD is its richness of natural compounds; in fact, a lot of MD foods contain numerous compounds that differ in their chemical structures and biological activities. In particular, some MD food products, such as tomato, extra virgin olive oil, hot pepper, red wine, vegetables (e.g., broccoli), milk thistle and various fruits, are motley rich in different types of compounds, such as phenols, flavonoids, terpenes, sterols, vitamins and coumarins. Furthermore, numerous studies have shown that these compounds can have different biological properties, such as antioxidant and anti-inflammatory activities, antihypertensive effects, lipid-lowering agents and anti-diabetic and anti-obesity effects. This means that even a single MD food product can improve several pathological conditions; thus, MD can have an important role in the prevention and therapy for different kinds of diseases, such as metabolic disorders, cancer and neurodegenerative diseases [33,34]. Tomato is one of the main food products of MD; its beneficial effects are an interesting object of study in metabolic disorders, especially the effects of lycopene (LYC), its principal component.

## 3. Lycopene

LYC, also known as γ, γ-carotene, is a liposoluble pigment that gives orange coloration to some vegetables and fruits (Figure 1). LYC is largely found in tomatoes, but it is also present in non-red and non-orange vegetable plants, such as asparagus and parsley [35,36]. LYC belongs to the carotenoid family and it is exclusively composed of carbon and hydrogen atoms and is a tetraterpene compound. LYC is characterized by an acyclic open-chain structure consisting of 8 isoprene units and 13 double bonds, 2 of which are unconjugated and 11 of which are conjugated forming a polyene chain [35,37]. This polyene chain primarily influences the chemistry and orientation of LYC within lipid bilayers [38]. The main feature of polyene chain functions as a light-absorbing chromophore that is responsible for the color of the food [39]. The 13 double bonds of LYC can interact with singlet oxygen and also with a wide range of other radicals, such as peroxyl radicals and, therefore, LYC can have antioxidant activity. When LYC interacts with radicals, the double bond can be cleaved or added interrupting the polyene chain and LYC can be degraded and lose its color [38,40]. Simple hydrocarbon carotenoids, as LYC is, show a strong tendency to micro-crystallization (forming aggregates) and some of the reported reactions with radicals can be influenced by such aggregation, especially where high concentrations of LYC are present. LYC is highly hydrophobic and, as a result, orientates itself deep within the lipophilic core of lipid bilayers [38].

LYC bioavailability is affected by food processing and cooking, dietary composition, mastication and by isomeric configurations [41,42]. Indeed, LYC exists in several forms and each of them is characterized by specific stability and bioavailability. Generally, LYC trans isomers are the predominant forms (approximately > 90%). Moreover, in LYC, all 13 double bonds are subjected to isomerization, resulting in differed cis isomers, such as 5, 9, 13 and 15, which can be observed in plants and blood plasma [43,44]. The LYC trans form represents the principal form present in fruit, but in cooked and processed products, such as tomato paste, this could fall to 35% [38]. Although trans isomers are the most diffuse, they have a lower bioavailability than the cis isomers [44]. In addition, the LYC 5-cis and all-trans isomers are the most energetically stable forms. Some cis isomers are recognized as having a better solubility than the all-trans form. Furthermore, some studies reported that the higher antioxidant activity of the cis isomers could be due to this improved solubility and with a lower tendency to self-aggregate in polar media. Indeed, it was suggested that the peroxyl radical scavenging ability of the LYC isomers was in the order 5-cis > 9-cis > 13-cis > all-trans. Most likely, this could be attributed to the improved solubility of cis isomers compared to the all-trans form [38]. On the contrary, LYC cis isomers in tissue and plasma are the most typical forms, which means that the largest part of the intake of LYC trans is converted into cis isomers through isomerization in enterocytes, liver and stomach [45,46]. Thus, the bioavailability and, consequently, the amount of LYC absorbed are improved [47,48]. Once LYC has reached the stomach, it is released from the food matrix by the mechanical churning and action of enzymes and acid. LYC can then be internalized into the lipid droplets and released into the small intestine where enzymes and bile acids continue the breakdown of the food matrix and where LYC can be taken up by enterocytes [49,50]. The absorption of LYC in the intestine happens because of passive diffusion and aided by scavenger receptor CD36 and B1 [51,52]. LYC could be partially cleaved by β-carotene oxygenase 1 (BCO1) and BCO2 into the enterocytes, but the majority of LYC is packed in chylomicrons. Subsequently, enterocytes release chylomicrons in the lymph and then into the portal circulation. In the liver, LYC is released from chylomicrons, which can be accumulated into low-density lipoproteins (LDL) or metabolized by liver enzymes, mainly by the cytochrome P450 [53,54]. Thus, absorbed LYC is distributed via the circulatory system with LDL accumulating in various tissues, preferentially in the testicles, adrenal glands, liver and prostate [55,56]. The LYC metabolites are formed by enzymatic or oxidative cleavage; some of the possible LYC products with bioactive properties include apo-lycopenals, apo-carotenedials, apo-lycopenones, carboxylic acids and epoxides [56,57].

## 4. LYC and NAFLD

LYC has various beneficial effects on human health and these beneficial activities have paved the way for the hypothesis that LYC could provide benefits in NAFLD patients. In a study conducted by Fenny et al., in mice, it was demonstrated that LYC decreases serum levels of TG and non-esterified fatty acids (NEFA) and attenuates liver steatosis with a decreased expression of lipogenic genes, such as acetyl-CoA carboxylase-1 (ACC1), fatty acid synthase (FAS) and sterol regulatory element-binding protein 1c (SREBP-1c) (Figure 2) [58]. LYC is a potent antioxidant agent; it acts on free radicals, such as ROS, hydrogen peroxide, nitrogen dioxide and hydroxyl radicals [56]. Furthermore, LYC can improve the cellular antioxidant defense system through the regeneration of the non-enzymatic antioxidants, vitamins E and C from their radicals. LYC induces the expression of cellular antioxidant enzymes, such as superoxide dismutase-1 (SOD1) and catalase (CAT) [59,60]. LYC can also enhance the activities of antioxidant enzymes in the liver as glutathione peroxidase (GPx) and antioxidant/detoxifying Phase II enzymes, such as heme oxygenase (HO-1) and glutathione S-transferase (GST), while the activity of ROS-producing enzymes, such as NADPH oxidase, inducible nitric oxide synthase (iNOS), cycloxygenase-2 (COX-2), 5-lipoxygenase (5-LOX) and CYP2E1 are reduced [61,62,63,64]. The anti-inflammatory activity of LYC is also important to prevent the development and the progression of NAFLD. Several studies have shown that LYC induced the expression of peroxisome proliferator activated-γ (PPARγ), reducing the production of pro-inflammatory cytokines [65]. The production of pro-inflammatory cytokines through the nuclear factor kappa-B (NF-κB) and mitogen-activated protein kinase (MAPK) signaling pathways is a crucial point in NASH progression [66,67]. Several studies have shown that LYC treatment leads to decreased MAPK, c-Jun N-terminal kinase (JNK) and NF-κB signaling activation associated with a reduced production of tumor necrosis factor-α (TNF-α) and interleukin-1β (IL-1β) and LPS-stimulated macrophage migration [68,69,70].

Several studies have shown that the white adipose tissue is an endocrine organ capable of releasing numerous adipokines and pro-inflammatory factors. It has been shown that high levels of adipokines, including IL-6 and TNF-α, can contribute to the reduction in lipid oxidation leading to lipotoxicity and insulin resistance. TNF-α is a key modulator of adipocyte metabolism owing to its direct role in glucose homeostasis and lipid metabolism. In addition, both IL-6 and TNF-α promote the production of leptin by adipose tissue, with an increase in inflammatory cytokines and the induction of the absorption of cholesterol by macrophages. In a study conducted by Luvizzotto et al., supplementation with LYC was observed to reduce leptin levels and restore plasma IL-6 concentrations significantly in obese rats [71].

Moreover, some LYC metabolites are bioactive, particularly apo-10′-lycopenoic acid (ALA), the major metabolite of LYC known to suppress hepatic steatosis and inflammation by stimulating sirtuin 1 (SIRT1). A study conducted by Chung et al. showed that ALA supplementation increased hepatic SIRT1 protein with a concomitant increase in the deacetylation of NF-κB p65, which in turn decreased the hepatic protein levels of IL-6 and TNFα by a mechanism not yet known [72]. The production of bioactive metabolites initially raised some doubts that the biological activities of LYC depended exclusively on its metabolites. A preclinical study conducted by Li et al. demonstrated that the antioxidant and anti-inflammatory activities that reduced steatosis in animals were mainly carried out by LYC [73]. Among carotenoids, we have found, in addition to lycopene, the substances astaxanthin, lutein and zeaxanthin. Astaxanthin is a carotenoid that is found in aquatic animals, such as salmon, trout, shrimp and lobsters, whist lutein is found in green leafy vegetables and zeaxanthin is found in corn [74]. Several studies conducted on mouse models have analyzed the protective effects of these substances in NAFLD. These studies have shown that astaxanthin, lutein and zeaxanthin share antioxidant activity. In this regard, in a study conducted by Xu et al. on mouse models with fatty liver, it was shown that astaxanthin caused an increase in the activity of SOD, CAT and GPx and a reduction in lipid peroxidation in the liver [75]. Undoubtedly, among the carotenoids, LYC has been at the center of the greatest scientific interest. In fact, the data regarding the other carotenoids in the treatment of NAFLD are currently limited.

## 5. Preclinical and Clinical Studies on the Relationship between LYC and NAFLD

Several studies have been carried out to verify the effects of LYC on NAFLD (Table 1). In a study conducted by Zidani et al., the effects of supplementing with 9% or 12% dry tomato peel (DTP) in mice fed a HFD for 12 weeks were analyzed [76]. The results of the study indicated that DTP decreased the plasma concentrations of LDL, TG, AST, ALT and alkaline phosphatase (ALP) in a dose-dependent manner. The beneficial effect of DTP, indicated by insulin concentration, plasma glucose and HOMA-IR, was significantly greater with 9% DTP compared with 12% DTP. Moreover, it has been observed that DTP increased the mRNA expression of adiponectin in the white adipose tissue. In a study by Róvero Costa et al., 24 rats were divided into four groups: control group, control group supplemented with LYC, obese group and obese group supplemented with LYC [77]. The main results shows the anti-inflammatory and antioxidant activity of LYC. In fact, LYC decreased the levels of TNF-α and IL-6, reducing the inflammatory response and, consequently, the extension of liver injury. Regarding the antioxidant activity, the LYC increased SOD and CAT activities. Finally, the lipid profile was analyzed, and this analysis showed decreased serum and hepatic TG levels and an increased HDL serum level in the obese group supplemented with LYC compared with the obese group (*p* < 0.05). In a study conducted by Jiang et al., 65 rats with NAFLD were divided into a control group and three groups treated with 5, 10 and 20 mg with LYC, respectively [69]. After the LYC treatment for six weeks, the body weight gain in the 10 and 20 mg/kg LYC-treated groups was lower compared to the model group (*p* < 0.01). Furthermore, the serum levels of AST and ALT were significantly decreased in a dose-dependent manner after the LYC treatment. Regarding the serum lipid levels, a remarkable reduction in the concentrations of LDL-C and TG in dose-dependent manners was observed, showing the lipid-lowering effects of LYC. Additionally, this study reported a significant increase in SOD and glutathione (GSH) levels in rats treated with LYC. Finally, the results of this study showed the decreased expression of TNF-α and CYP2E1, involving a reduction in inflammatory response [69]. In a study conducted by Piña-Zentella et al., the effects of LYC in rats with NAFLD were examined [78]. After inducing NAFLD with a HFD, the rats were divided into three groups: the control group, the group fed with a normal diet and the group fed with a normal diet supplemented with LYC (20 mg/kg). The results of the study indicated that, after four weeks in the group supplemented with LYC, the liver weight was reduced (*p* = 0.002), and serum LDL levels (*p* = 0.008) and TC concentrations in the liver were normalized (*p* = 0.001). Regarding oxidative stress enzymes, it was observed that SOD and CAT activities were increased in the group on a normal diet supplemented with LYC. The macroscopic analysis found a complete recovery of injured livers. In a study conducted by Negri et al., 61 obese children with fatty liver were enrolled; all participants underwent a washout low carotenoid diet for two weeks, and then they were randomly assigned to a restricted calorie regimen (RCR) (Group 1) or RCR supplemented with tomato juice and basil and oregano extracts in extra virgin olive oil (Group 2) for 60 days [79]. Subsequently, the patients were switched to the alternative regimen for the next 60 days. Numerous improvements were identified in both groups when LYC was introduced into a diet, but, in group 2, they were more pronounced. The study reported that the body weight decreased significantly (*p* < 0.01) accompanied with the loss of subcutaneous and visceral fat and decreased liver size. Additionally, the serum levels of insulin decreased by 34% in group 2. Total cholesterol showed a decrease in group 2 of 6.7%. Regarding the LYC antioxidant activity, it was detected that GSH levels increased in both groups after being exposed to LYC. Instead, the anti-inflammatory activity was ascertained by the decrease in IL-4 and increase in adiponectin.

## 6. Effects of LYC on Fibrosis and HCC

NASH is closely associated with hepatic fibrosis and, if not adequately treated, can evolve into HCC [80,81]. The antioxidant activity of LYC would appear to have an important role in NASH as it reduces the concentrations of ROS produced in hepatocytes [56]. In the study conducted by Ni et al., the effects of LYC on fibrosis resolution and NASH inversion were analyzed [82]. During the study, the mice were divided into three groups: a control group fed a diet rich in cholesterol and fatty acids (CL) and two groups fed with CL added at different concentrations of LYC (0.004% and 0.012%). The results of the study showed that the concentration of LYC at 0.012% reduced the plasma levels of TG, AST and ALT. In the second phase of this study, a second group of mice were divided and treated with normal diet (NC), NC and LYC, only CL and CL and LYC. The study results showed increased levels of hydroxyproline and smooth muscle alpha-actin (αSMA) (marker for fibrotic activity) in mice fed with CL, while the groups fed with LYC showed a reduction in hydroxy-proline and αSMA levels compared to the CL group (*p* < 0.01 and *p* < 0.05, respectively). Another study by Wang et al. showed that LYC can inhibit hepatocarcinogenesis induced by NASH [83]. In particular, the rats were divided into of six groups: normal diet (ND) group (as the control group), ND and LYC group, CD and tomato extract (TE) group, HFD group, HFD and LYC group, HFD and TE group. After six weeks, the authors observed a reduction of 80% in impaired hepatic foci in the HFD and LYC and HFD and TE group compared to the HFD group (*p* < 0.01), suggesting that both supplementations have anticarcinogenic actions. In addition, a reduction in hepatocytes, which exposed the proliferating cell nuclear antigen (PCNA), was observed in all groups where LYC or tomato extract was administered (*p* < 0.05), indicating that LYC could inhibit cell proliferation induced by NASH. Since oxidative stress contributes to the onset of liver cancer, the levels of certain cellular components involved in oxidative stress were analyzed. In particular, it was noted that the HFD and LYC group showed an increased expression of transcription factor 2 related to nuclear factor E2 (Nrf2), a key regulator of cellular response to oxidative stress (*p* < 0.05). Nrf2 is the main regulator of the enzyme heme oxygenase-1 (HO-1) that acts as an antioxidant and eliminates free radicals [84]. The ability to increase the expression of Nrf2, therefore, gives LYC hepatoprotective properties as it prevents the formation of cancer cells. In the study conducted by Ip et al., the chemoprotective activity of LYC was analyzed [85]. The purpose of this study was to demonstrate the inhibitory effects of LYC on cancer genesis, and if these effects were directly due to LYC or its metabolites. Therefore, a wild-type (WT) strain of mice and a BCO2-knockout strain (BCO2-KO) were used. The BCO2-knockout strain was characterized by the absence of the genes encoding the BCO2 enzyme responsible for converting LYC into its metabolites. In this study, four groups of mice fed a HDF were observed for 24 weeks: WT and BOC2-KO group as control, WT and LYC supplementation group and BCO2-KO and LYC supplementation group. After 24 weeks, the result of the study indicated a reduction in the incidence of HCC in both groups fed with LYC with a reduction of 17% for the group WT and LYC and 20% for the group BCO2-KO and LYC. Finally, another study conducted by Ip et al. showed that the ALA supplementation in mice for 24 weeks significantly reduced hepatic tumorigenesis HFD induced (50% reduction in tumor multiplicity; 65% in volume) [86]. ALA seems to reduce hepatic tumorigenesis by stimulating SIRT1 signaling, while reducing hepatic inflammation. Currently, no clinical data are available in the literature analyzing the effects of LYC in HCC.

## 7. Conclusions

Currently, NAFLD presents a new challenge for scholars and physicians. A balanced and healthy diet with reduced amounts of saturated fat is the strategy that gives beneficial effects both in the prevention and in the treatment of NAFLD. Several studies are available in the literature that confirm the effectiveness of the MD in improving NAFLD owing to a great variety of nutraceutical components, among which LYC is one of the most abundant components. Several LYC biological functions in the body can be particularly effective against a multifactorial pathology, such as NAFLD. Particular attention is paid to the antioxidant activity, which is already known to act efficiently in NAFLD. In fact, many studies have reported an increase in the activity of SOD and CAT as a final result of the administration of LYC. The anti-inflammatory activity of LYC has also been confirmed in several studies performed for the reduction in TNF-α and CYP1E2 expressions. Further, the reduction in serum levels of lipids and hepatic levels of cholesterol by LYC was confirmed, indicating a possible action on the lipid metabolism. The mechanism for these results has not yet been understood fully. Many preclinical studies are available in the literature with the aim to clarify the mechanism of LYC action. However, there is a great lack of clinical studies. In conclusion, we can state that LYC currently presents a very promising natural compound in the prevention and treatment of NAFLD, but more studies by the scientific communities are certainly indispensable to investigate some of its effects in a more useful way. A large multicenter clinical study is needed to confirm the efficacy of LYC in the treatment of NAFLD.

## Figures and Tables

**Figure 1 jcm-11-03477-f001:**
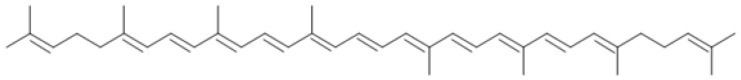
Chemical structure of LYC.

**Figure 2 jcm-11-03477-f002:**
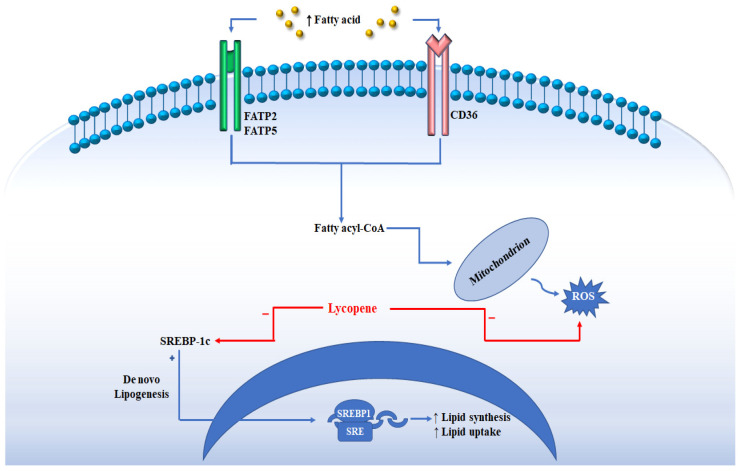
The figure shows how the uptake of circulating fatty acids is facilitated by the transporters present on the membrane FATP2/FATP5 (fatty acid transport protein 2 or 5), and CD36 (cluster of differentiation 36). Once inside the cell, the fatty acids can undergo the beta oxidation process with their transport to the mitochondrion. During NAFLD, the excess of fatty acids determines an increase in the production of ROS. The de novo lipogenesis regulated by SREBP-1c contributes to the accumulation of fat in the liver. The figure indicates how lycopene interacts by reducing the levels of SREBP-1c and ROS.

**Table 1 jcm-11-03477-t001:** Effects of LYC on NAFLD.

Study		Nutritional Protocol	Results	Ref.
Jiang et al., 2016	Wistar rats	High-fat diet and LYC (5, 10 or 20 mg)	↓ AST, ALT, TG, LDL TNF-α levels and CYP2E1 expression ↑ HDL, SOD and GSH	[69]
Piña-Zentella et al., 2016	Sprague Dawley rats	High-fat diet, followed by normal diet with LYC	↓ LDL ↑ SOD and CAT activity	[78]
Zidani et al., 2017	mice	High-fat diet and DTP with 9% and 17% of LYC	↓ LDL-C, TG, AST, ALT and ALK levels	[76]
Róvero Costa et al., 2019	Wistar rats	Hypercaloric diet and LYC	↑ HDL levels, SOD and CAT activity ↓ TG and TNF-á levels	[77]
Negri et al., 2020	Obese children	Restricted calorie regimen and tomato juice	↓ Body weight, ALT, AST, LDL, IL4, TG and insulin levels, ↑ HDL, GSH and adiponectine levels	[79]

## Data Availability

Not applicable.

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
