# Peer review of "Mediterranean Diet: The Beneficial Effects of Lycopene in Non-Alcoholic Fatty Liver Disease"

_jcm, 2022, doi:10.3390/jcm11123477_

Round 1
Reviewer 1 Report
The authors looked at a study of non-alcoholic fatty liver disease (NAFLD), which is the most common liver disease, and the carotenoid lycopene, which can alleviate the effects of this disease. The main risk factors are components of the metabolic syndrome such as insulin resistance, type II diabetes and prediabetes, abdominal obesity and hypercholesterolaemia. A high-calorie, high-carbohydrate diet (especially fructose and glucose-fructose syrup), processed foods and fast food is also beneficial for the disease. With the development of the obesity epidemic, an increasing incidence of fatty liver that is not caused by alcohol consumption is observed.
Other causes that may be responsible for fatty liver include:
• certain medicines such as methotrexate, amiodarone, estrogens, steroids, sodium valproate, warfarin,
• exposure to certain chemicals, e. g. chloride derivatives of hydrocarbons;
• parenteral nutrition,
• conditions after removal of a part of the intestine,
• Diseases of the pancreas
• Crohn’s disease and ulcerative colitis,
• certain congenital metabolic diseases,
• Viral hepatitis,
• Pregnancy
The authors of the Mediterranean Diet (MD) dedicated a chapter to alleviate the effects of NAFLD. Many studies suggest the role of a low-calorie diet and the choice of healthy foods as prevention of NAFLD. Many studies point to a Mediterranean diet. This diet is characterized by a high consumption of vegetables, fruits and nuts, olive oil, fish and dairy products. The health properties are due to the presence of fiber, vitamins and polyphenols. It is recommended that people with NAFLD eat a low-calorie, individually tailored diet. The energy supply should be 1200-1500 kcal/day; 25-30 kcal/kg body weight, based on the set weight or addition of 20-40% to the basal metabolic rate. A deficit of 500 to 1000 kcal should lead to a weight loss of 0. 5 to 1. 0 kg per week. Various studies have shown that dietary intervention is accompanied by personalized physical training, significantly reduces the fat content of the liver or insulin immunity in overweight and obese people, regardless of weight loss.
Specific comments:
The work title should be reviewed as a large section of the article is devoted to the Mediterranean diet and its impact on the NAFLD.
It is worth mentioning the intake of broccoli, soy phospholipids and milk thistle seeds in the diet.
Chapter 5 on the preclinical and clinical research of lycopene and its effects on NAFLD is well written. However, there are no studies in adult subjects.
Chapter 6 on the effect of lycopene on fibrosis and HCC.
Clinical trials should be included in this chapter. After editing and completing the text, the article will meet the requirements of the J. Clin Med.
Author Response
The authors looked at a study of non-alcoholic fatty liver disease (NAFLD), which is the most common liver disease, and the carotenoid lycopene, which can alleviate the effects of this disease. The main risk factors are components of the metabolic syndrome such as insulin resistance, type II diabetes and prediabetes, abdominal obesity and hypercholesterolaemia. A high-calorie, high-carbohydrate diet (especially fructose and glucose-fructose syrup), processed foods and fast food is also beneficial for the disease. With the development of the obesity epidemic, an increasing incidence of fatty liver that is not caused by alcohol consumption is observed.
Other causes that may be responsible for fatty liver include:
- certain medicines such as methotrexate, amiodarone, estrogens, steroids, sodium valproate, warfarin,
- exposure to certain chemicals, e. g. chloride derivatives of hydrocarbons;
- parenteral nutrition,
- conditions after removal of a part of the intestine,
- Diseases of the pancreas
- Crohn’s disease and ulcerative colitis,
- certain congenital metabolic diseases,
- Viral hepatitis,
- Pregnancy
The authors of the Mediterranean Diet (MD) dedicated a chapter to alleviate the effects of NAFLD. Many studies suggest the role of a low-calorie diet and the choice of healthy foods as prevention of NAFLD. Many studies point to a Mediterranean diet. This diet is characterized by a high consumption of vegetables, fruits and nuts, olive oil, fish and dairy products. The health properties are due to the presence of fiber, vitamins and polyphenols. It is recommended that people with NAFLD eat a low-calorie, individually tailored diet. The energy supply should be 1200-1500 kcal/day; 25-30 kcal/kg body weight, based on the set weight or addition of 20-40% to the basal metabolic rate. A deficit of 500 to 1000 kcal should lead to a weight loss of 0. 5 to 1. 0 kg per week. Various studies have shown that dietary intervention is accompanied by personalized physical training, significantly reduces the fat content of the liver or insulin immunity in overweight and obese people, regardless of weight loss.
Specific comments:
The work title should be reviewed as a large section of the article is devoted to the Mediterranean diet and its impact on the NAFLD.
Thanks for your comment. The text was changed according to the comment.
It is worth mentioning the intake of broccoli, soy phospholipids and milk thistle seeds in the diet.
Thanks for your comment. The text was changed according to the comment.
Chapter 5 on the preclinical and clinical research of lycopene and its effects on NAFLD is well written. However, there are no studies in adult subjects.
Thanks for your comment. Currently, the data in the literature are limited
Chapter 6 on the effect of lycopene on fibrosis and HCC. Clinical trials should be included in this chapter.
Thanks for your comment. Currently, the data in the literature are limited.
After editing and completing the text, the article will meet the requirements of the J. Clin Med.
Thanks for your comment
Reviewer 2 Report
In this manuscript, Ludovico Abenavoli et al made a comprehensive review of “Beneficial effects of Lycopene in Non-alcoholic fatty liver disease” (NAFLD). This review provides an overview of lycopene metabolism, molecular pathway and the potential beneficial effect of lycopene for NAFLD. I have some comments and questions as below:
1. According to the topic of this manuscript, the role and importance of Lycopene should be mentioned in the section of Mediterranean diet. Vegetables, not only tomato, but also bean and nuts, are included in the Mediterranean diet. Is the benefit effect of Mediterranean diet for NAFLD from the lycopene?
2. Could author make a brief comparison between lycopene and other carotenoids for patients with NAFLD? In clinical practices, patients with NAFLD frequently ask about the protective effect of different carotenoids, such as astaxanthin, lycopene, β-carotene and lutein against the development and progression of NAFLD.
3. Adipokines, such as adiponectin and leptin, are important in the mechanism for the development of NAFLD. Could authors add information about the relationship between lycopene and adipokines?
4. In line 230-240, authors mentioned sirtuin 1 (SIRT1) could suppress hepatic steatosis and inflammation. Apo-10′-lycopenoic acid (ALA) supplementation increased hepatic SIRT1 protein with a concomitant increase in deacetylation of NF-κB p65. Some studies (such as Blanche C. Ip, nutrition 2014) also reported the anti-tumor effect, especially inhibit tumorigenesis and metastasis of hepatocellular carcinoma (HCC). Please add this information of SIRT1 effect on the section “6. Effects of LYC on fibrosis and HCC”.
5. Minor correction: (A) please re-adjust the sequence of references in table 1 according to years (2016 to 2020). (B) in text, “P” value should be “P” (italic).
Author Response
In this manuscript, Ludovico Abenavoli et al made a comprehensive review of “Beneficial effects of Lycopene in Non-alcoholic fatty liver disease” (NAFLD). This review provides an overview of lycopene metabolism, molecular pathway and the potential beneficial effect of lycopene for NAFLD. I have some comments and questions as below:
- According to the topic of this manuscript, the role and importance of Lycopene should be mentioned in the section of Mediterranean diet. Vegetables, not only tomato, but also bean and nuts, are included in the Mediterranean diet. Is the benefit effect of Mediterranean diet for NAFLD from the lycopene?
Thanks for your question. The Mediterranean diet is rich in compounds with activity in NAFLD which have been largely summarized in other works, the authors' attention has been focused on Lycopene as this carotenoid presents a highly interesting profile that the authors have tried to enhance.
- Could author make a brief comparison between lycopene and other carotenoids for patients with NAFLD? In clinical practices, patients with NAFLD frequently ask about the protective effect of different carotenoids, such as astaxanthin, lycopene, β-carotene and lutein against the development and progression of NAFLD.
Thanks for your comment. The text was changed according to the comment.
- Adipokines, such as adiponectin and leptin, are important in the mechanism for the development of NAFLD. Could authors add information about the relationship between lycopene and adipokines?
Thanks for your comment. The text was changed according to the comment
- In line 230-240, authors mentioned sirtuin 1 (SIRT1) could suppress hepatic steatosis and inflammation. Apo-10′-lycopenoic acid (ALA) supplementation increased hepatic SIRT1 protein with a concomitant increase in deacetylation of NF-κB p65. Some studies (such as Blanche C. Ip, nutrition 2014) also reported the anti-tumor effect, especially inhibit tumorigenesis and metastasis of hepatocellular carcinoma (HCC). Please add this information of SIRT1 effect on the section “6. Effects of LYC on fibrosis and HCC”.
Thanks for your comment. The text was changed according to the comment
- Minor correction: (A) please re-adjust the sequence of references in table 1 according to years (2016 to 2020). (B) in text, “P” value should be “P” (italic).
Thanks for your comment. The text was changed according to the comment